# Pharmacological inhibition of α-synuclein aggregation within liquid condensates

Samuel T. Dada[1], Zenon Toprakcioglu[1], Mariana P. Cali [1], Alexander Röntgen[1], Maarten C. Hardenberg[1], Owen M. Morris[1], Lena K. Mrugalla[1], Tuomas P. J. Knowles [1] & Michele Vendruscolo [1] ✉

Aggregated forms of α-synuclein constitute the major component of Lewy bodies, the proteinaceous aggregates characteristic of Parkinson's disease. Emerging evidence suggests that α-synuclein aggregation may occur within liquid condensates formed through phase separation. This mechanism of aggregation creates new challenges and opportunities for drug discovery for Parkinson's disease, which is otherwise still incurable. Here we show that the condensation-driven aggregation pathway of α-synuclein can be inhibited using small molecules. We report that the aminosterol claramine stabilizes α-synuclein condensates and inhibits α-synuclein aggregation within the condensates both in vitro and in a *Caenorhabditis elegans* model of Parkinson's disease. By using a chemical kinetics approach, we show that the mechanism of action of claramine is to inhibit primary nucleation within the condensates. These results illustrate a possible therapeutic route based on the inhibition of protein aggregation within condensates, a phenomenon likely to be relevant in other neurodegenerative disorders.

The aggregation of α-synuclein within dopaminergic neurons of the substantia nigra plays a critical role in the onset and progression of Parkinson's disease (PD), a neurodegenerative condition that is still incurable and results in motor disorders, cognitive deficits, and autonomic dysfunctions[1,2]. Although the aggregation of α-synuclein has been extensively studied, its links with the early events that lead to pathological processes in PD remain not well established. This incomplete mechanistic understanding poses a challenge when developing therapeutic strategies to eliminate or to reduce the agents of neurotoxicity[3]. A generally accepted mechanism by which α-synuclein aggregates takes place a through pathway, known as the deposition pathway, where the protein transitions from the native state to amyloid state through a nucleation and growth process[4,5]. Promising advances have been made to target this pathway for pharmacological interventions[3,6–10]. Recent evidence, however, suggests that the amyloid state can also be formed through another pathway, known as the condensation pathway, where the native state converts first into a liquid-like condensed state through a phase separation process,

and then aggregates into the cross-β structure characteristic of the amyloid state[11–16].

Protein phase separation has been intensively studied in the last decade because of its possible links with the formation of biomolecular condensates, which have been suggested to play important biological roles in the cell[17–19]. It has also been realized that the high concentrations of protein within the liquid-like condensates create the conditions for their conversion into solid-like amyloid fibrils[20,21], including tau[22], a protein that forms neurofibrillary tangles in Alzheimer's disease, FUS[23], hnRNPA1[24] and TDP-43[25], three proteins whose aggregation has been linked to amyotrophic lateral sclerosis (ALS), and α-synuclein[11–13,26]. Therefore, it is of great interest to investigate whether it is possible to prevent this conversion using pharmacological interventions[27,28]. One route to achieve this result is to identify small molecules capable of stabilizing the liquid-like condensates with respect to the solid-like aggregates, as it was recently reported for tau[29–31].

Since the kinetic network that describes the mechanism of aggregation of α-synuclein is the same along the deposition and condensation pathways[26] (Supplementary Fig. 1), one may explore

[1]Department of Chemistry, Centre for Misfolding Disease, University of Cambridge, Cambridge CB2 1EW, UK. ✉e-mail: mv245@cam.ac.uk

whether compounds reported to inhibit the aggregation of α-synuclein along the deposition pathway[6–10,32] may also work along the condensation pathway. However, because the concentration in the condensates is up to three orders of magnitude higher than in the dilute phase[26], compounds that inhibit α-synuclein aggregation along the deposition pathway may lose their potency in the condensates[33].

To investigate how small molecules can modulate the phase behaviour of α-synuclein, we studied claramine[34], a spermine-containing aminosterol (Fig. 1A) structurally similar to trodusquemine and squalamine[35]. These two natural products were initially isolated from the liver of dogfish sharks, and shown to cross the blood-brain barrier and possess antimicrobial, antiangiogenic, and antitumor properties which has made this family of compounds of

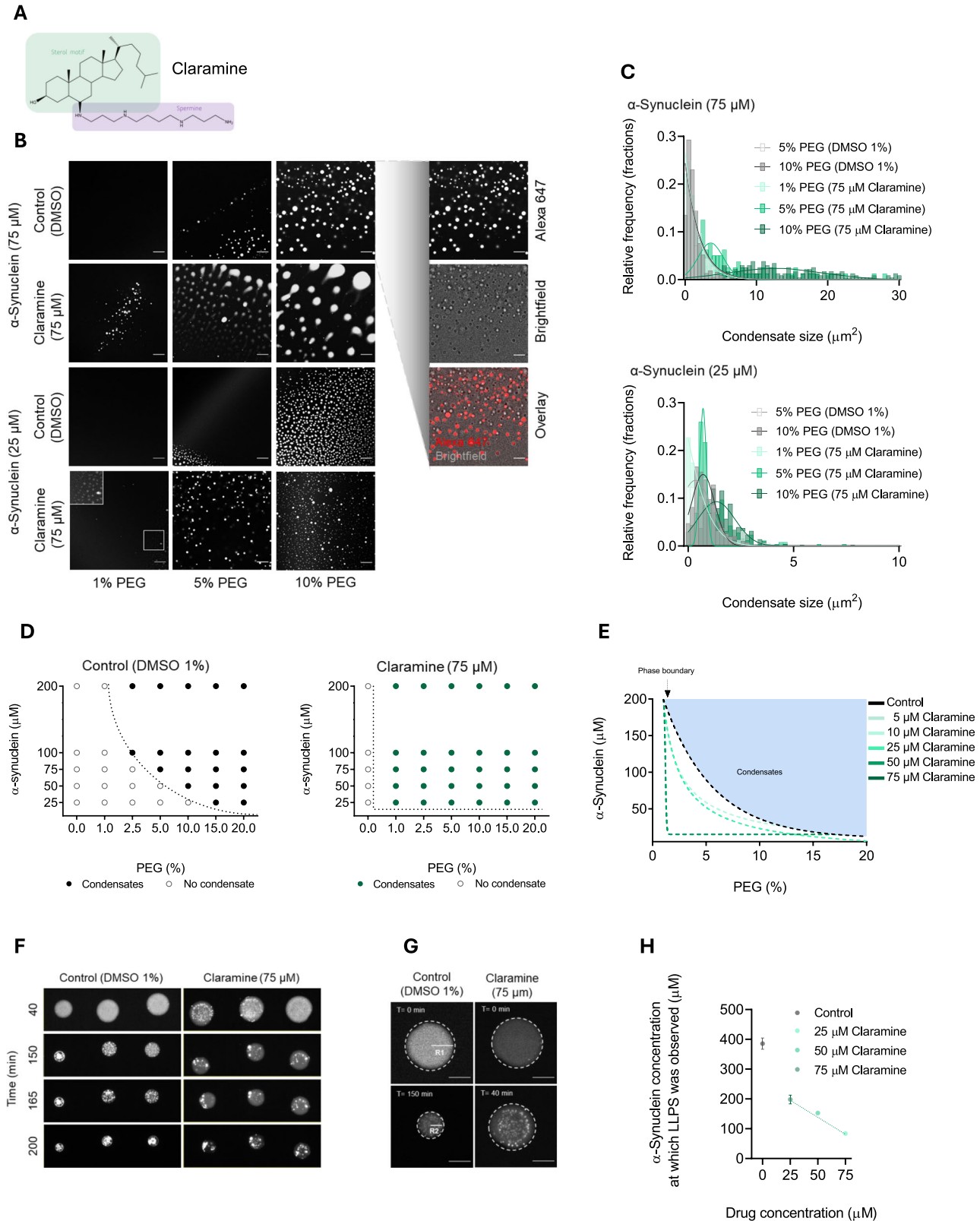

**Fig. 1 | Claramine increases the propensity of synuclein to undergo phase separation. A** Chemical structure of claramine showing the spermine side chain (purple) and the sterol group (green). **B** Representative fluorescence images of α-synuclein condensate formation in the presence and absence of claramine (75 μM) at different PEG concentrations. Images were obtained 10 min post incubation. The scale bar represents 5 μm. **C** Histogram of the size distribution and relative frequency of condensates for the images displayed in panel (**B**) at 5% (off-white),10% (grey) PEG in the absence of claramine, and 1% (light green), 5% (green) and 10% (dark green) PEG in the presence of claramine (75 μM). The bin width for conditions with 75 μM and 25 μM α-synuclein are 0.5 and 0.2, respectively. **D** Phase diagram for different PEG and α-synuclein concentrations (1% DMSO) in the absence (left graph) and presence of claramine (75 μM) (right graph) at which phase separation was observed after a 10 min incubation period. The dots indicate the tested conditions, where hollow dots indicate lack of phase separation, whilst solid dots indicate phase separation. The dotted black line represents the phase boundary. **E** Phase diagram for different α-synuclein and PEG concentrations with the addition of different claramine concentrations. The phase boundary for the different conditions tested are represented by the dotted lines; increasing concentrations of claramine are highlighted by the green increasing gradient colour. **F** Representative fluorescence images displaying condensate formation of six droplets trapped within a microfluidic chamber overtime. **G** Enlarged images showing droplet (1% DMSO) in the absence and presence of claramine (75 μM). The scale bar represents 50 μm. **H** Concentration of α-synuclein at which phase separation was observed within droplets as shown in panels F and G, in the absence (grey) and presence (green) of different concentrations of claramine, at 25 (light green), 50 (green) and 75 (dark green) μM, in 10% PEG. All experiments were performed in 50 mM Tris-HCl at pH 7.4 in the presence of 10% PEG unless otherwise stated. Data shown are representative of experiments repeated at least three times. Results are mean ± SEM.

therapeutic importance[36,37]. Additionally, both trodusquemine and squalamine have been reported to inhibit α-synuclein aggregation by respectively, displacing oligomeric α-synuclein from the cell membrane and protecting neuronal cells from oligomer toxicity[8,35,38]. Squalamine has recently entered Phase 2b human clinical trials for the treatment of PD, as an orally administered phosphate salt known as ENT-01[3].

Although claramine has only been reported on its anti-microbial properties and anti-diabetic effects in its ability to reduce the cleavage of insulin receptors[34,39], it has been suggested that this aminosterol could have the capacity to be beneficial to treat neurodegenerative diseases[35,39,40]. Following the recent establishment of the aggregation mechanism of α-synuclein within condensates, and the determination of the corresponding kinetic rate constants, using a fluorescence-based aggregation assay[26], we first investigated whether claramine could modulate the phase separation of α-synuclein. We then explored the effects of claramine on the kinetic rate constants when the aggregation is accelerated within condensates. We also evaluated the implications of claramine administration to the physiology and liquid-like properties of α-synuclein inclusions in a well-established transgenic *Caenorhabditis elegans* (*C. elegans*) model of PD which expresses α-synuclein inclusions in body wall muscle cells[13,41].

## Results

### Claramine stabilizes α-synuclein condensates

Claramine is a small cationic molecule containing a polyamine (spermine) covalently bound to a fused sterol ring (Fig. 1A). To systematically characterise the effects of claramine on the condensate formation of α-synuclein, we employed an approach that involves a confocal microscopy-based assay and a microfluidic device to monitor protein phase separation[26,42]. Microscopy was conducted by adding a small sample drop on a microscopy glass slide, whilst microfluidics required the injection and trapping of a small sample into a chamber enclosed by fluorinated oil for observation. As protein phase separation is typically driven by low-affinity interactions, these assays have been shown to be effective in detecting changes in α-synuclein phase separation behaviour in the presence of sodium chloride and 1,6-hexanediol, which are well-known modulators of α-synuclein protein phase separation through electrostatic and hydrophobic interactions, respectively[11–13,26].

First, we aimed to evaluate the effects that claramine has on altering the propensity of α-synuclein (25–200 μM) to phase separate at physiological pH using polyethylene glycol (PEG), a molecular crowding agent[11,26], at various concentrations. Overall, condensates were observed to be visibly larger with increasing protein (75 μM) and PEG (10%) concentrations (Fig. 1B, C). However, the sizes of the condensates were enhanced in the presence of claramine at all tested concentration of α-synuclein and PEG when compared with those formed in the absence of claramine (Fig. 1B, C). All observations were made in the first 10 min from the moment the sample was deposited onto the glass slide.

Given these findings, we next assessed how claramine influences the phase boundary separating the dilute and condense states of α-synuclein. We found that the presence of claramine shifts the phase boundary towards the condense state in a concentration-dependent manner, when charaterising the condense state as a function of α-synuclein and PEG concentration (Fig. 1D, E and Supplementary Fig. 2). At high concentrations of claramine (50 and 75 μM) as little as 1% PEG was required to induce the phase separation of α-synuclein (Fig. 1D, E and Supplementary Fig. 2). Using a microfluidic device, we were able to trap the microdroplets, monitor them over time and observe them as they reduce in size whilst condensates form. This procedure enabled us to estimate the concentration of α-synuclein required to drive its phase separation (Fig. 1F, G). We noticed that the addition of claramine reduced the concentration of α-synuclein needed for phase separation to occur, as the concentration was calculated to be an average of 386, 198, 152 and 84 μM for DMSO (1%), 25, 50 and 75 μM claramine respectively (Fig. 1H). As a control we examined the impact of the spermine backbone solely on the condensate formation of α-synuclein and spermine did not alter the propensity of α-synuclein to undergo phase separation (Supplementary Fig. 3). Taken together, these results demonstrate that claramine increases the tendency of α-synuclein to phase separate. Claramine, therefore, stabilizes the condensate state of α-synuclein, while not modifying it covalently or binding to α-synuclein strongly as claramine displayed a dissociation constant ($K_D$) of about 20 μM (Fig. 1B–H and Supplementary Fig. 4). Given that phase separation is known to rely on low-affinity electrostatic interactions, and an increase in ionic strength has been linked to a higher yield of amyloid fibrils in α-synuclein, we chose to investigate the influence of adding claramine on α-synuclein phase separation in a high ionic environment[11,13,26]. The introduction of claramine was observed to visibly enhance the ripening process of α-synuclein under high ionic conditions of 150 mM NaCl (Supplementary Fig. 5). Additionally, FRAP measurements at the 15 min time point revealed about 80% fluorescence recovery within α-synuclein condensates in the presence of claramine (Supplementary Fig. 6). In contrast, absence of claramine resulted in only ~45% fluorescence recovery at the same time point (Supplementary Fig. 6). However, at the 10 min time point no significant differences in the rate of recovery was observed between the two conditions as elongation of α-synuclein within the condensates has just begun (Supplementary Fig. 6B).

### Claramine inhibits α-synuclein aggregation within condensates

Next, we studied whether or not claramine could affect α-synuclein amyloid formation from within α-synuclein condensates. Other aminosterols (trodusquamine and squalamine) have been shown to inhibit the protein aggregation of α-synuclein monomers in a lipid-induced aggregation assay[8,35,38]. Therefore, we tested whether

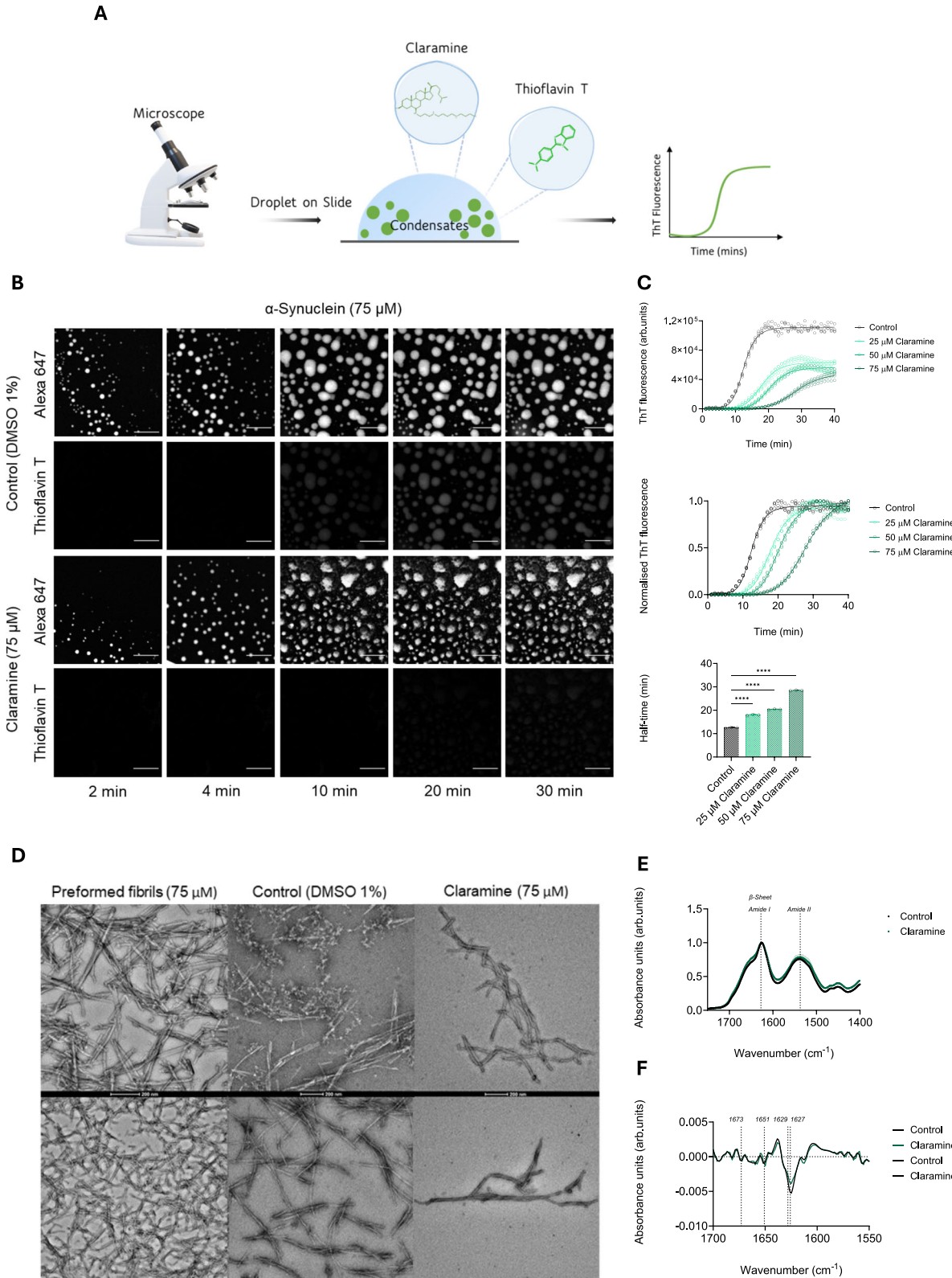

claramine had a similar impact on α-synuclein using a thioflavin-T (ThT) strategy for studying α-synuclein aggregation within liquid condensates[26] (Fig. 2A). Firstly, we ensured that claramine did not interfere with the fluorescence signal of ThT. Additionally, we confirmed that claramine did not quench the ThT signal in the aggregation assay (Supplementary Fig. 7). We monitored the aggregation of monomeric α-synuclein at a concentration of 75 μM in 50 mM Tris-HCl at pH 7.4 in the presence of 10% PEG and 20 μM ThT at room temperature (-23 ± 2 °C), using confocal microscopy in the absence and presence of claramine at three different concentrations (25, 50 and 75 μM). The results revealed that claramine reduced the β-sheet formation of α-synuclein observed by the low ThT signal observed in comparison to the control (Fig. 2B). A quantitative analysis validated this observation, as claramine reduced the amyloid load, aggregation

**Fig. 2 | Claramine slows down the aggregation of α-synuclein within condensates. A** Schematic diagram illustrating the components of the thioflavin T (ThT) based assay used to monitor aggregation within condensates. The buffer system was 50 mM Tris-HCl at pH 7.4, 10% PEG and monomeric α-synuclein labelled was labelled with Alexa Fluor 647 for visualization. **B** Fluorescence images showing α-synuclein condensate formation and aggregation in the absence (1% DMSO) and presence of claramine (75 μM) over time. The images represent an area of sample tracked over time; the scale bar represents 20 μm. **C** Quantification of ThT emission for images shown in panel (**B**) for 75 μM α-synuclein in the presence of 1% DMSO (control) (black), 25 μM (light green), 50 μM (green) and 75 μM (dark green) claramine over a 40 min time period. The top graph displays the ThT emission, the middle graph shows the normalised ThT emission, and the bottom graph highlights the corresponding aggregation half-times for 75 μM α-synuclein in the presence and absence of claramine. **D** Representative transmission electron microscopy (TEM) images of preformed α-synuclein fibrils (75 μM), and of α-synuclein fibrils (75 μM) post phase separation (>40 min) in the absence (1% DMSO) and presence of claramine (75 μM). The scale bar represents 200 nm (upper panel) and 500 nm (lower panel), respectively. **E** Fourier-transform infrared (FTIR) spectra of recovered products from the α-synuclein (75 μM) phase separation assay displaying the amide I and amide II regions in the absence (1% DMSO) (black) and presence of claramine (75 μM) (dark green). **F** Second derivative FTIR spectra of deconvoluted amide I region from panel (**E**) showing the band frequency assignments assigned to structures post phase separation of α-synuclein (75 μM) in the absence (1% DMSO) (black) and presence of claramine (75 μM) (dark green). All experiments were performed using 75 μM α-synuclein in 50 mM Tris-HCl at pH 7.4 in the presence of 10% PEG unless otherwise stated. The data represent the mean ± SEM of $n = 4$ individual experiments. A one-way ANOVA test with Dunnett's multiple comparisons correction was used in panel (**C**) (****$P < 0.0001$).

rates and half-time of α-synuclein aggregation in a concentration-dependent manner (Fig. 2C). The data that we obtained suggest that claramine strongly modulates α-synuclein amyloid formation within condensates by controlling the rate of primary nucleation (with rate constant $k_n$).

We next sought to determine and characterise the morphology and structure of the amyloid aggregates formed post phase separation in the absence and presence of claramine by transmission electron microscopy (TEM) and Fourier transform infrared (FTIR) spectroscopy (Fig. 2D–F). TEM images displayed no visible differences in the α-synuclein fibrillar structures formed in the absence and presence of claramine. However, the number of fibrils was lower in the presence of claramine (Fig. 2D). We then used FTIR spectroscopy to investigate whether there were any structural differences in the β-sheets present in the α-synuclein fibrils formed in the absence and presence of claramine (Fig. 2E, F). We found that the presence of claramine at an equimolar concentration to α-synuclein did not significantly impact the secondary structure composition of the protein aggregates, as both samples had amide I and II bands at around 1650 cm⁻¹ and 1540 cm⁻¹ respectively, which is characteristic of a random coil structure (Fig. 2E). However, the intensity of parallel (1629 cm⁻¹ and 1627 cm⁻¹) and not the antiparallel (1,673 cm⁻¹) β-sheet content increased in the presence of claramine as observed by the normalised secondary derivative analysis (Fig. 2F). Furthermore, we carried out some biochemical analysis and evaluated the influence of claramine on the soluble and insoluble fractions obtained from the phase separation aggregation assay. We observed a concentration-dependent rise in the levels of monomeric α-synuclein in the soluble fraction with an increase in claramine concentration (Supplementary Fig. 8). Conversely, the blot revealed an inverse relationship, indicating that higher claramine concentrations corresponded to decreased concentrations of oligomeric and aggregated α-synuclein in the insoluble fraction (Supplementary Fig. 8). This further supports the observed reduction in ThT amplitude in the presence of Claramine in Fig. 2C.

## Claramine only mildly influences the seeded aggregation of α-synuclein within condensates

To probe further the mechanism of action of claramine on α-synuclein aggregation within condensates, we sought to bypass the primary nucleation step through the addition of preformed fibrils of α-synuclein (Fig. 3A). This approach enabled us to assess the effects of claramine on secondary processes, as the preformed fibrils promote the growth and amplification of the fibrils[5,26,43]. This happens because the surfaces of the preformed fibrils act as catalytic sites for promoting secondary nucleation (with rate constant $k_2$) and as elongation sites (with rate constant $k_+$).

We thus monitored the aggregation of α-synuclein within condensates in the presence of 2% preformed fibrils (monomer equivalents) and claramine at ratios of 3:1 (25 μM) 3:2 (50 μM) and 1:1 (75 μM) of α-synuclein (75 μM) to claramine. In the presence of small amounts of seeds (2%), the rate of fibril formation is driven by secondary nucleation and elongation[5,26,43]. We observed that claramine was effective at reducing the fibrillar load of α-synuclein within condensates at higher concentrations, but less effective at a lower concentration of 25 μM in the presence of 2% seeds (Fig. 3B, C). However, the addition of the 2% preformed fibril reduced the dependence of aggregation kinetics on the α-synuclein to claramine ratio, and this dependency is diminished at higher concentrations of claramine where the half-time of 50 μM is >75 μM claramine (Fig. 3C). In the presence of 25% preformed fibrils primary and secondary nucleation are bypassed, and elongation of the preformed fibrils becomes the dominant process of aggregate growth[5,26,43]. We found that addition of claramine was able to only mildly reduce the fibrillar yield (Fig. 3D, E). Additionally, the presence of 25% preformed fibrils eliminated the concentration dependent effect of claramine on the aggregation of α-synuclein within condensates (Fig. 3E). These findings further indicate that claramine only has a small effect on secondary nucleation and elongation of α-synuclein aggregation within condensates.

To quantify the effect of claramine on the aggregation of α-synuclein within condensates, we employed a chemical kinetics framework that enables us to translate the aggregation profiles in terms of the rate constants for the individual contributing microscopic steps (primary nucleation, fibril elongation and secondary nucleation)[43]. Using the web-based software AmyloFit, we characterised the complex reaction networks associated with protein aggregation using experimentally obtained kinetic data[43]. The fitting was first carried for the control (without claramine) of unseeded and seeded kinetic data and then for samples prepared using three different concentrations of claramine by globally fitting a single-step nucleation model to the kinetic traces (Supplementary Fig. 9). We found the fitting to be well described by the rate constants involving primary nucleation, thus, all parameters for $k_+k_2$ obtained through the analysis were kept constant as the rate of primary nucleation was shown to decrease by an order of magnitude with increasing concentrations of claramine (Table S1). This finding therefore suggests that claramine modulates the aggregation of α-synuclein within condensates by controlling the rate of primary nucleation. These results are in accordance with what has been observed with other aminosterols, which have been shown to inhibit the lipid-induced aggregation of α-synuclein[8,38].

## Claramine reduces the formation of α-synuclein assemblies, increases the liquidity of α-synuclein inclusions and fosters the fitness in a *C. elegans* model of PD

To determine the effects of claramine in vivo we used a well-established transgenic *C. elegans* model of PD, in which α-synuclein, fused to a yellow fluorescent protein (YFP), is overexpressed in the muscle cells of their body[13,41]. As the nematode worms age, α-synuclein accumulates forming inclusion bodies, leading to physiological dysfunctions which can be phenotypically observed by monitoring their

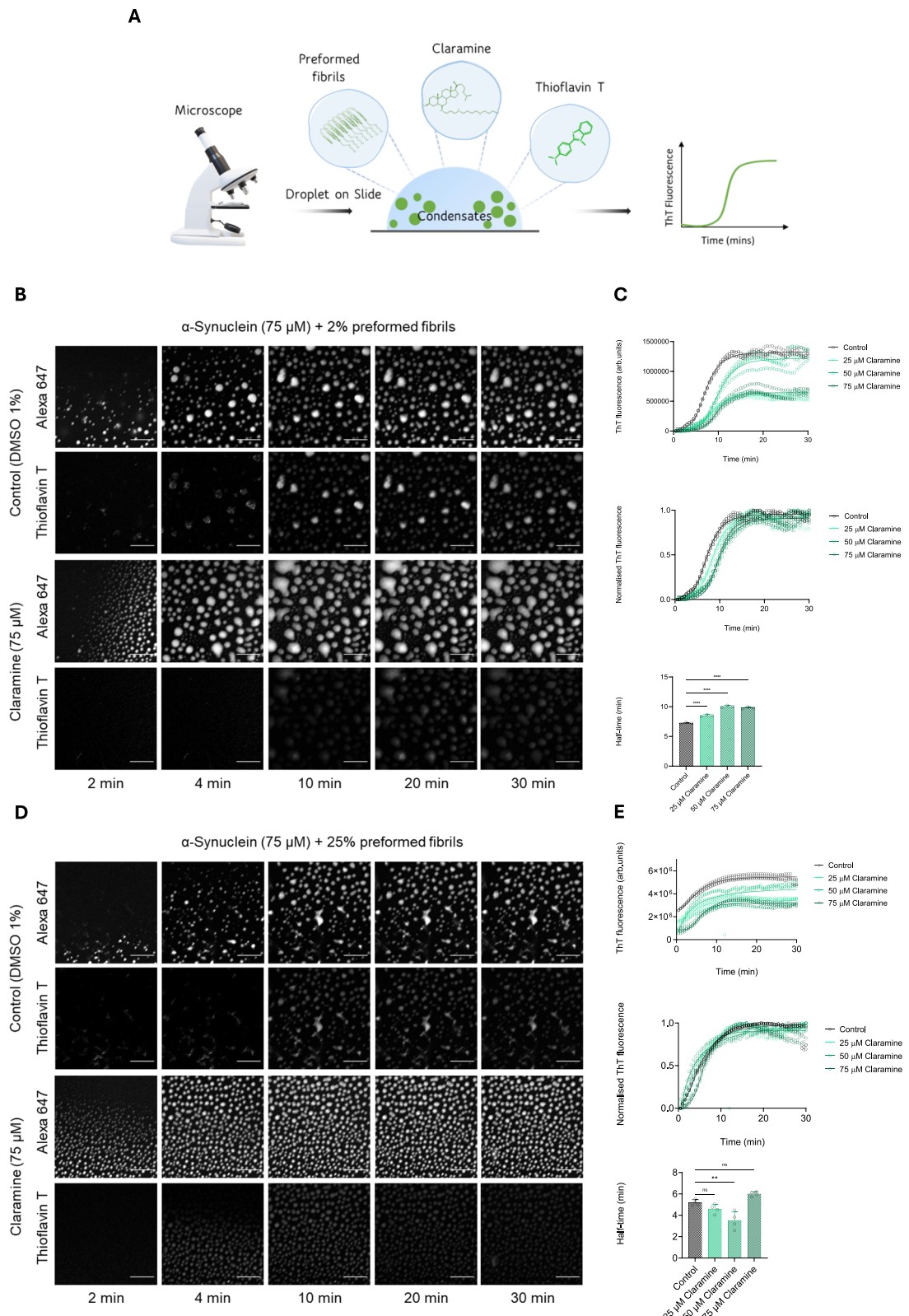

**A**

**B** α-Synuclein (75 μM) + 2% preformed fibrils

**C**

**D** α-Synuclein (75 μM) + 25% preformed fibrils

**E**

motility in units of bends per minute (BPM) and inclusion number over time using a nematode tracking platform and confocal microscope respectively[44]. Claramine at 5 μM was administered to the nematodes by its addition on top of the nematode growth media, seeded with bacteria at the L4 larval stage of development, prior to adulthood as α-synuclein aggregates are only visible thereafter. This concentration of claramine was selected based on our experiments with human

neuroblastoma SH-SY5Y cells, which revealed that concentrations exceeding 10 μM of Claramine were toxic to the cells (Supplementary Fig. 10A). Additionally, claramine did not demonstrate the ability to suppress the toxicity of α-synuclein fibrils when compared to the control (Supplementary Fig. 10B).

We first sought to assess the aggregation of α-synuclein in the nematodes with and without the addition of claramine from day 4 of

**Fig. 3 | Claramine mildly affects the aggregation of α-synuclein within condensates in the presence of preformed fibril seeds. A** Schematic diagram illustrating the components of the ThT based assay used to monitor aggregation within liquid condensates. The components of this assay mimics that of the assay described in Fig. 2A with the addition of 2% or 25% preformed fibrils. **B–E** Representative fluorescence imaging displaying α-synuclein condensate formation and aggregation with 2% preformed fibrils (**B**) and 25% preformed fibrils (**D**), in the absence (1% DMSO) and presence of claramine (75 μM) over time. The images represent an area of sample tracked over time; the scale bar represents 20 μm. ThT emission quantification of the 2% and 25% seeded assay images shown in panels (**B**) and (**D**) respectively, for 75 μM α-synuclein in the presence of 1% DMSO (control)

(black), 25 μM (light green), 50 μM (green) and 75 μM (dark green) claramine over a 30 min time period. The top graph on both (**C**) and (**E**) displays the raw ThT values, the middle graphs show the normalised kinetic profile of the aggregation assay, and the bottom graphs shows the corresponding half-times of aggregation for 75 μM α-synuclein in the presence and absence of claramine. All experiments were performed using 75 μM α-synuclein with either 2% or 25% preformed fibrils in 50 mM Tris-HCl at pH 7.4 in the presence of 10% PEG unless otherwise stated. The data represent the mean ± SEM of $n = 4$ individual experiments. A one-way ANOVA test with Dunnett's multiple comparisons correction was used in (**C**) and (**E**) (n.s –not significant, **$P < 0.01$, ***$P < 0.001$, ****$P < 0.0001$).

adulthood. We observed an exponential increase in the number of α-synuclein inclusions from day 4–15 of adulthood (Fig. 4A, B). However, nematodes administered claramine had a reduced number of inclusions when compared with the control treated α-synuclein-YFP nematodes on day 7, 11 and 15 of adulthood (Fig. 4A, B). By contrast, the administration of claramine did not lead to the formation of visible inclusions or negative physiological implications on the control worm strain, which only expressed YFP without α-synuclein in the muscle cells of their body, as their YFP expression pattern was not visually observed to be affected by the administration of claramine (Fig. 4A and Supplementary Fig. 11A–C).

We then investigated the implications of claramine administration on the motility of the nematodes by using an automated nematode tracking platform to quantitatively characterize the physiological behaviour of the nematodes in a meticulous manner[44]. This phenotypic charaterisation method has been used extensively for distinguishing and identifying genes and molecular pathways implicated in age-related protein homoeostasis and neurotoxicity[44]. The functional phenotypic assessment of the α-synuclein-YFP nematodes was found to correlate with the observed effects on the formation of α-synuclein inclusions (Fig. 4C). This is because the α-synuclein-YFP nematodes treated with claramine had a higher average bend per minute than the control treated nematodes on all four tested days of adulthood (Fig. 4C and Supplementary Fig. 11D). Although we observed a greater difference in the average bends per minutes on day 11 rather than any other days, the results highlight that claramine appears to share similarities in its mode of action with compounds that inhibit the aggregation process[8,38]. On day 4 of adulthood, the toxicity levels of the α-synuclein aggregates are low, hence the lack of significant difference in the motility of the nematodes treated with claramine against the control (Fig. 4C). Then, as toxicity builds up on day 7, as a result of ageing and aggregation of α-synuclein, the control nematodes were more affected than the nematodes treated with claramine, due to its delaying aggregation effects (Fig. 4C). On day 11, this effect was observed more prominently, as motility in claramine treated nematodes was significantly higher, and eventually, as claramine treatment wears off, the motility rate becomes equal to the control as observed on day 15 (Fig. 4C and Supplementary Fig. 11D). This finding demonstrates the protective effect of claramine in recovering the motility and decreasing the number of inclusion bodies.

We observed an age-dependent coalescence of the α-synuclein-YFP inclusions in the transgenic nematodes, as previously reported[13,41]. We then investigated the dynamics and liquid-like properties of the α-synuclein assemblies by performing fluorescence recovery after photobleaching (FRAP) within the nematodes (Fig. 4D, E). α-Synuclein inclusions were found to be dynamic and liquid-like on day 4 of adulthood, as indicated by the fast recovery after photobleaching, with the fluorescence recovery of α-synuclein inclusions in nematodes treated with claramine being about 40% higher than the control (Fig. 4E). Ageing of the nematodes, however, resulted in an overall decline in the fluorescence recovery (~ 20%) as seen on day 15, which confirms a liquid to solid-like transition (Fig. 4D, E). It has been previously reported that a-synuclein inclusions between days 1 and 11 of

adulthood, are predominantly non-amyloid like, however, inclusions in nematodes aged days 13–15 of adulthood appear to be amyloid-like[13]. Remarkably, in our study, on day 15 of adulthood, inclusions in nematodes treated with claramine were observed to have around 40% fluorescence recovery after photobleaching, which reflects the low-affinity interactions in inclusion assembly and further proves that claramine modulates α-synuclein to thermodynamically favour the droplet/liquid-like state its aggregation pathway (Fig. 4D, E).

## Discussion
In this study we have explored the possibility of a drug discovery strategy to inhibit the aggregation of α-synuclein within liquid-like condensates. We have used an in vitro ThT-based aggregation assay that enables the real-time observation of amyloid formation in a reliable manner. We have revealed that the aminosterol claramine is a strong modulator of α-synuclein phase separation by reducing the free energy barrier required for α-synuclein to transition from the monomeric state to the condensed state, whilst also increasing the free energy barrier for the conversion of α-synuclein from the condensed state to the amyloid state in vitro. We have also shown that this molecule inhibits α-synuclein aggregation by reducing the rate of the primary nucleation microscopic step of α-synuclein aggregation. Furthermore, these observations paralleled the in vivo studies, as the protective effect of the exogenous administration of claramine was shown by the decrease in the number of inclusions, minor recovery of motility and reduced maturation rates of inclusions in a well-studied *C. elegans* model of PD. Overall, the modulating activity of claramine on α-synuclein aggregation in vitro and in vivo under phase separated conditions, strongly corroborates the outcomes obtained by other aminosterols like trodusquemine and squalamine. Due the encouraging results of squalamine in first phase of clinical trials for PD, the results of this study illustrate a potential avenue to prevent aggregation of misfolded α-synuclein within condensates.

## Methods
### Chemicals
Claramine and spermine (Sigma-Aldrich, MO, USA) were synthesised as a trifluoroacetate salt (for claramine) at a purity >98% as measured by high-performance liquid chromatography (HPLC), stored as a lyophilized powder, and solubilised in DMSO (100%) to a final concentration of 10 mM. Molecules were stored at −80 °C and thawed once before each experiment.

### Expression and purification of α-synuclein
The wild-type and cysteine (A90C) variants of α-synuclein were expressed using *E. coli* BL21 (DE3)-gold competent cells (Agilent Technologies) expressing the pT7-7 plasmid encoding α-synuclein. The expression of α-synuclein was induced by the addition of 1 mM isopropyl β-d-1-thiogalactopyranoside (IPTG). Cells were subsequently centrifuged, lysed by sonication, denatured by boiling before the precipitation of α-synuclein using ammonium sulphate. Following this, protein pellets were dialysed and purified through ion exchange and size exclusion chromatography into 50 mM trisaminomethane-

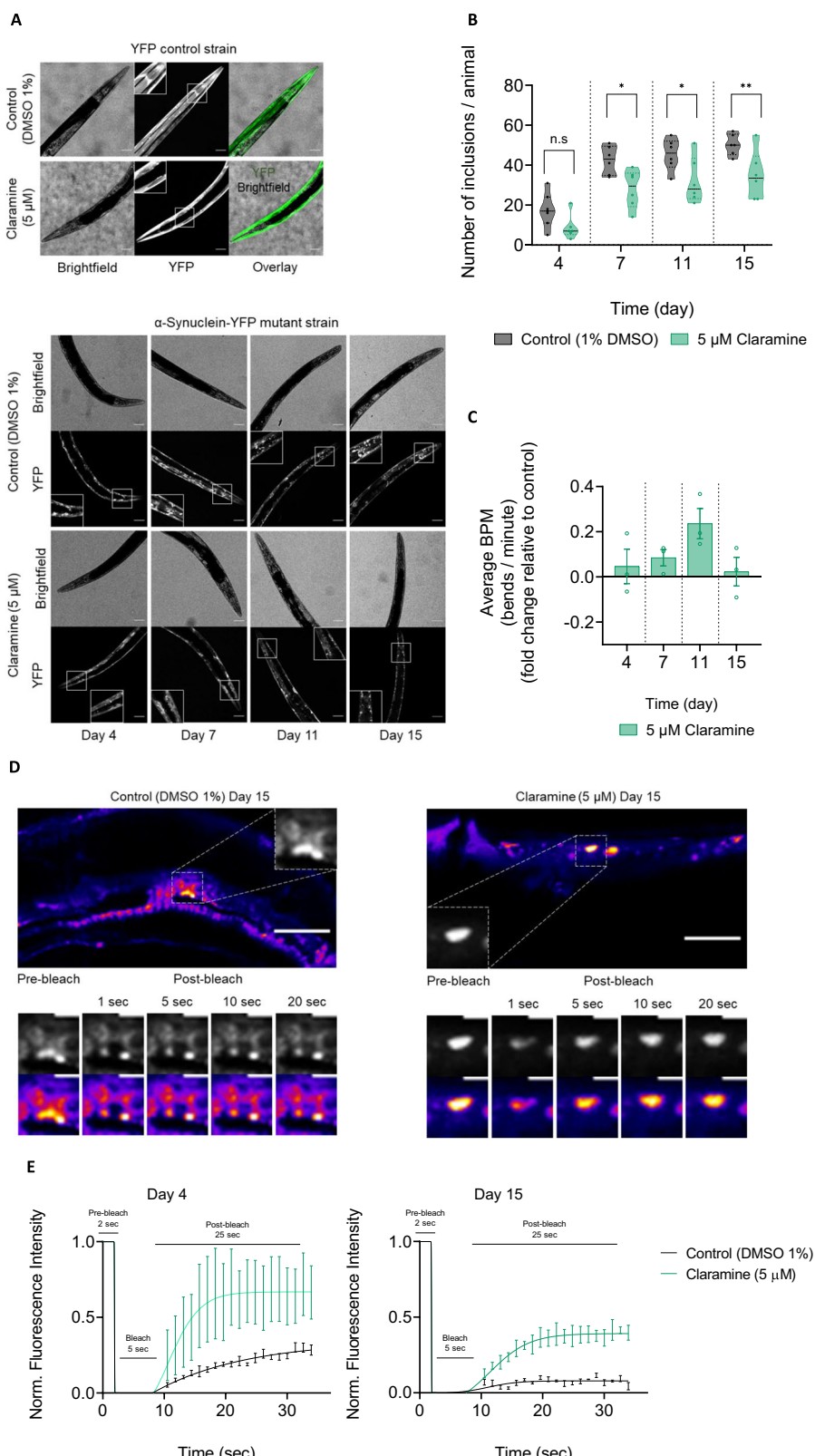

hydrochloride (Tris-HCl) pH 7.4 buffer. All buffers used in the dialysis and purification of the α-synuclein A90C cysteine variant contained 1 mM dithiothreitol (DTT) to prevent the formation of disulfide bonds. The final protein concentration was measured using ultraviolet-visible (UV-vis) spectroscopy on a Cary 100 system (Agilent Technologies). All proteins were aliquoted, flash-frozen in liquid nitrogen, stored at −80 °C and thawed once before each experiment.

### α-Synuclein labelling

The A90C α-synuclein was labelled with 1.5-fold molar excess of C5 maleimide-linked Alexa Fluor 647 (Invitrogen Life Technologies) overnight at 4 °C under constant gentle stirring. The unbound dye was removed using Amicon Ultra-15 Centrifugal Filter Units and buffer exchanged into 50 mM Tris-HCl at pH 7.4 by size exclusion chromatography. The final protein concentration was measured using

**Fig. 4 | Claramine reduces the formation and maturation of α-synuclein inclusions, as well as muscle paralysis in a *C. elegans* model of PD.**
**A** Representative images showing the effects of claramine (5 µM) in both the YFP control strain and the α-synuclein-YFP PD mutant strain. The top panel shows that 1% DMSO or 5 µM claramine did not have significant effects on the on the YFP expression of the YFP control strain on day 7 of adulthood. The bottom panel shows the progression of the inclusion assembly in the body wall muscle cells over time (1% DMSO) between days 4 and 15 of adulthood in the absence and presence of claramine (5 µM). The scale bar represents 20 µm. **B** Quantification of images shown in panel A (bottom) of α-synuclein-YFP inclusions in PD worms at indicated time points (1% DMSO) in the absence (grey) and presence of 5 µM claramine (green). At least six worms were analysed in total. **C** Data from an automated worm motility assay showing the relative fold change in average bends per minute of worms

treated with 5 µM claramine over time between days 4 and 15 of adulthood. At least 50 worms were analysed in total per experiment ($n = 3$). **D** FRAP images of α-synuclein-YFP inclusions on day 15 of adulthood. Images on the left of the panel corresponds to representative worm administered DMSO (1%) whilst the panel on the right corresponds to worm administered 5 µM claramine. The images correspond to the region of interest with pre-bleach and post-bleach droplets at 1, 5, 10 and 20 s for each tested condition. The scale bar represents 20 µm and 5 µm for the top and bottom images respectively. **E** Normalised recovery traces from FRAP experiment for α-synuclein-YFP inclusions treated with DMSO (1%) (black) and 5 µM claramine (green) on day 4 and day 15 of adulthood, at least 4 worms were analysed in total per condition. The data represent the mean ± SEM. A two-way ANOVA test with Sidak's multiple comparisons correction was used in **B** (n.s –not significant, *$P < 0.1$, **$P < 0.01$).

ultraviolet-visible (UV-vis) spectroscopy on a Cary 100 system (Agilent Technologies). All proteins were aliquoted, flash-frozen in liquid nitrogen, stored at −80 °C and thawed once before each experiment.

### α-Synuclein phase separation assay
To induce condensate formation, non-labelled wild-type α-synuclein was mixed with the A90C variant labelled with Alexa Fluor 647 at a 100:1 molar ratio in either 50 mM Tris-HCl pH 7.4 and 10% polyethylene glycol 10,000 (PEG) (Thermo Fisher Scientific) by volume at room temperature (20–22 °C). Additionally, claramine, dimethyl sulfoxide (1% DMSO), spermine, NaCl and preformed fibrils were added to the protein phase separation assay for various experiments. 10 µL of the final mixture was pipetted on a 35 mm glass bottom dish (P35G-1.5-20-C, MatTek Life Sciences) and immediately imaged on a Leica Stellaris Will inverted confocal microscope using a 40×/1.3 HC PL Apo CS2 oil objective (Leica Microsystems). The excitation wavelength was set to 633 nm for all experiments. All images were processed and analysed in ImageJ (NIH).

### α-Synuclein aggregation assay within condensates
Thioflavin T (ThT) 20 µM (Sigma), claramine, 1% DMSO and preformed fibrils, depending on the experiments, were mixed with monomeric wild-type α-synuclein, containing 1 molar % A90C α-synuclein labelled with Alexa Fluor 647 prior to each experiment. The assay mixture, which included 50 mM Tris-HCl, pH 7.4, 10% PEG 10,000, was pipetted onto a 35 mm glass bottom dish and imaged on a Leica Stellaris Will inverted confocal microscope using a 40 × /1.3 HC PL Apo CS2 oil objective. The excitation wavelength 633 nm and 488 nm were used for Alexa Fluor 647 labelled α-synuclein and ThT, respectively. Images were acquired every min for ~40 min. Images were processed on ImageJ: an area adjacent to the edge of the droplet was cropped and analysed, thus allowing for time-dependent measurement of amyloid formation at the droplets.

### α-Synuclein preformed fibril seeds preparation
α-Synuclein pre-formed fibrils were formed from recombinant α-synuclein wild-type monomers diluted in buffer (50 mM Tris-HCl at pH 7.4) to concentrations of ~500 µM. Monomers in were incubated at 40 °C, with constant stirring speed (1,500 rpm) with a PTFE micro stirrer bar and left to aggregate on an RCT Basic Heat Plate (RCT Basic, model no. 0003810002; IKA, Staufen, Germany) for up to 72 h. Samples were centrifuged at 4 °C at 18,800 g for 15 min. Once fibrils were isolated from supernatant, they were suspended in buffer equivalent to discarded supernatant. The fibril concentration was measured by dissociating a small aliquot of fibrils in a total solution of 4 M guanidinium hydrochloride (GndHCl). After a 30 min incubation period, the fibril concentration was measured using UV-vis on a Cary 100 system (Agilent Technologies). The fibrils were aliquoted and stored at room temperature before experimentation. Before each experiment, fibrils were pre-treated by 15 s (15 pulses) sonication at 10% power with 50%

duty cycle using a Microtip sonicator (Bandelin Sonopuls HD2070) to disperse lumped fibrils.

### Fourier-transform infrared spectroscopy (FTIR)
Product from protein phase separation assay in the presence of DMSO (1%) and claramine (75 µM) upon condensate and amyloid formation were washed off glass slide with 50 mM Tris-HCL. Product was centrifuged at 13,800 g for 10 min, at room temperature. Supernatant containing soluble α-synuclein was discarded, and pellets were washed twice with $H_2O$. Pellets were resuspended in 3 µL $H_2O$ and subjected to attenuated total reflectance (ATR) Fourier transform infrared (FTIR) spectroscopy. 2 µL of solution were applied to a Perkin-Elmer Spectrum100 FTIR with an ATR diamond attachment and dried. Prior to scanning, sample and electronics chambers were purged with a constant flow of dry air. 100 replicate scans were averaged from 4000–800 cm$^{-1}$, normalized to amide I intensity (-1630 cm$^{-1}$ peak), and second derivatives were taken with 9 points for slope analysis.

### Estimation of the α-synuclein concentration required for phase separation in water-in-oil droplets
Fabrication of microfluidic devices: The fabrication process of the microfluidics was taken from a previously established protocol[42,45,46]. In brief, a soft photolithographic process was used to fabricate the master through which microfluidic devices were made. A 50 µm photoresist (SU-8 3050, MicroChem) was spin-coated onto a silicon wafer. This was soft baked at 95 °C for 3 min. A film mask was placed on the wafer and the whole system was exposed in UV light to induce polymerization. The wafer was then baked at 95 °C for 30 min. Finally, the master was placed into a solution of propylene glycol methyl ether acetate (PGMEA, Sigma-Aldrich), which helped in the development process. Elastomer polydimethylsiloxane (PDMS) with curing agent (Sylgard 184, DowCorning, Midland, MI) was mixed at a ratio of 10:1 to fabricate the devices. This mixture was then incubated at 65 °C and cured for a total of 3 h. Once hardened, the PDMS was peeled off the master, and holes were punched into the PDMS, which acted as inlets and outlets. Finally, the PDMS slab was bound to a glass slide by treatment with a plasma bonder (Diener Electronic, Ebhausen, Germany).

Formation and confinement of droplets: *neMESYS syringe pumps*. Syringe pumps (Cetoni, Korbussen, Germany) were used to control the flow rates within the microchannels. Protein solution was mixed with PEG at a ratio of 1:1 at the first junction. At the second junction, the oil phase, which consisted of fluorinated oil (Fluorinert FC-40, Sigma-Aldrich) and 2% w/w fluorosurfactant (RAN biotechnologies) intersected the aqueous phase resulting in water-in-oil droplets being formed. Following droplet generation, droplets were confined within a microfluidic trap[42,45,46]. In brief, droplets are directed towards an array of traps whereby once a droplet is driven within the microfluidic confinement, it is unable to escape unless a pressure is applied from the outlet. Droplets were then incubated at room temperature to allow for shrinkage. This resulted in an increase of the local concentration of

protein and PEG within the droplets which led to protein phase separation. The water-in-oil droplets and the protein phase separation was monitored using fluorescence microscopy.

Calculation of protein concentration during phase separation: The concentration of the protein was obtained by calculating the ratio of the droplet volume just after trapping ($V_1$) and at the point of phase separation ($V_2$), i.e. at the point at which condensates start appearing within the water-in-oil droplet. By multiplying the initial monomeric protein concertation by the value of this ratio, the actual protein concentration at the point of phase separation could be determined.

## Transmission electron microscopy (TEM)

α-Synuclein samples from the ThT-based aggregation assay that contained either DMSO (1%) or claramine (75 μM) were obtained after fibril formation. The obtained sample (5 μL) was deposited on a carbon film of 400 mesh 3 mm copper grid. The grids were washed once with Milli-Q water, then incubated with 1% (w/v) uranyl acetate for 2 min and washed twice again with Milli-Q water before being air-dried at room temperature. Samples were imaged using a Tecnai G2 transmission electron microscope operating at 80–200 keV.

## Liquid-chromatography mass spectrometry (LC-MS)

Monomeric α-synuclein was diluted in storage buffer to a concentration of 5 μM in the absence and presence of claramine (1:1 protein drug molar ratio). Protein liquid-chromatography mass spectrometry (LC-MS) was performed on a Xevo G2-S TOF mass spectrometer coupled to an Acquity UPC system using an Acquity UPLC BEH300 C4 column. The mobile phase was composed of $H_2O$ with 0.1% formic acid (solvent A) and 95% MeCN and $H_2O$ with 0.1% formic acid (solvent B) at a flow rate of 0.2 mL/min. The electrospray source was operating with a capillary and cone voltage of 20 kV and 40 C, respectively. Nitrogen was applied as desolvation gas at a total flow rate of 850 L/h. The mass spectra were reconstructed using the MaxEnt algorithm on the MassLynx software according to the manual.

## Microscale thermophoresis (MST)

α-Synuclein was spun down at 4 °C and 21,100 g for 20 min prior to the experiment. A serial dilution of 16 concentrations (15 μL per sample) of claramine was diluted into 50 mM Tris buffer (pH 7.4). To account for residual DMSO, the buffer was supplemented with DMSO accordingly. In addition, 0.2% Tween-20 was added to the buffer to prevent the sticking of the protein to the capillary walls. To each dilution sample, 5 μL of 1.5 μM monomeric α-synuclein (98% wild type; 2% Alexa Fluor 647-labelled) was added. The samples were incubated for 30 min before the MST measurement was performed. Samples were loaded onto the Monolith NT™ standard capillaries and placed in the sample tray of the Monolith NT.115 instrument. All thermophoresis experiments were performed at 22 °C, 30% red LED intensity and 50% infrared laser (IR) intensity with the laser being on for 30 s per capillary. All analysis was performed using the MO. Screening Analysis Software.

## Quenching assay

α-synuclein fibrils were prepared by incubating monomers in a 50 mM Tris-HCl pH 7.4 solution at a concentration of 345 μM. The sample were incubated in an Eppendorf tube under agitation at 1500 rpm at 40 °C for 72 h on a RCT Basic Heat Plate (RCT Basic, model no. 0003810002, IKA, Staufen, Germany). After incubation, solutions containing the fibrils were subjected to centrifugation at 4 °C, 18,800 g for 15 min. Following centrifugation, the supernatant was discarded, and the fibril precipitate was resuspended in the same volume of buffer to maintain a concentration of 345 μM (monomeric equivalents). The fibrils were sonicated for 15 pulses of 15 s at 10% power with 50% duty cycle using a Microtip sonicator (Bandelin Sonoplus HD2070) to disperse lumped fibrils prior to the experiment. 5 mM stock of claramine was prepared in DMSO and filtered using 0.02 μM syringe filters (Whatman Anotop

10/0.02). Solutions containing 75 μM (monomeric equivalents) of fibrils, 20 μM of ThT, and varying concentrations of claramine (25, 50, and 75 μM) were prepared. The fluorescence intensity of each solution was compared to a control solution consisting of 75 μM of fibrils and 20 μM of ThT. All samples were aliquoted into a 96-well plate, and ThT fluorescence was monitored for 30 min in a plate reader using an excitation wavelength of 440 nm and an emission wavelength of 480 nm. All experiments were conducted as biological replicates, and the data are presented as mean ± SD. Statistical significance between different experimental groups was analyzed using an one-way ANOVA test with multiple comparisons correction, with GraphPad Prism 10 (GraphPad Software) being used for all statistical analyses.

## Fluorescence recovery after photobleaching (FRAP)

FRAP experiments were performed on condensates using a Leica Stellaris Will inverted stage scanning confocal microscope. To conduct FRAP experiments, a 63 x magnification oil objective (63x/1.4 HC PL Apo CS oil) was used. Bleaching was done using the 647 nm laser at 20% intensity for 2 s following a 2 s pre-beach sequence. Immediate post-beach images were captured at a rate of 1 s per frame for 20 s. Intensity traces of bleached area were background-corrected and normalised to reference signal and FRAP time which is defined by the time to half maximal signal recovery. This was determined using the FRAP wizard system (Leica) on the confocal microscope.

## Western blot analysis

Phase separation assay was carried out in glass dish with and without claramine. Samples were extracted from glass dish with 50 mM Tris-HCl and centrifuged 21,100 g for 20 min to separate the soluble (found in sample supernatant) and insoluble (found in sample pellet) fractions, equal volume of buffer was added to insoluble fraction. Both soluble and insoluble (without boiling) samples were loaded onto a NuPAGE 12% Bis-Tris precast protein gel and ran for 35 min at 200 V. To transfer protein from the gel to a nitrocellulose membrane, an iBLOT2 dry blotting system was used. The membrane was blocked in PBS with 5% BSA at 4 °C overnight and then washed three times with PBS. The membrane was incubated with an Alexa Fluor 488 anti his tag primary antibody (488 anti-α-synuclein MJFR1 (Abcam, catalogue #ab195025); 1:5000 dilution in PBS with 5% BSA) for 3 h at room temperature, washed five times with PBS with 0.02% tween before imaging.

## AmyloFit data analysis

The aggregation of α-synuclein within droplets was monitored as described above. The experimental ThT fluorescence readout was uploaded on the free online platform AmyloFit[43] (https://www.amylofit.ch.cam.ac.uk). Next, the software normalised the data to 0% and 100% by averaging the values at the baseline and the plateau of the reaction. Upon data normalisation, the concentration of aggregate mass could be observed as a function of time. The minimum and maximum fluorescence intensity, and the aggregation half-time were calculated from the time at which half the protein that is present, initially as monomer, has aggregated, i.e., the time at which the normalized intensity reaches 0.5. A basin-hopping algorithm was applied to fit the experimental data to a model of protein aggregation. Each experiment was repeated four times and then averaged before fitting, while a primary nucleation dominated model was assumed. The number of basin hops was set to 40 for each fit. The applied model was only considered suitable if it was able to match the experimental data well. The AmyloFit user manual can be consulted for more in-depth information on the fitting procedure[43].

## Cell culture

Human SH-SY5Y neuroblastoma cells were cultured in DMEM/F-12, GlutaMAX (Thermo Fisher Scientific) with 10% foetal bovine serum

(FBS) on 75 cm² cell culture bottles (Greiner Bio-One) at 37 °C and 5% CO₂ and split at 80% confluency.

### MTT cell viability assay

Cell viability was measured using the 3-(4,5-dimethylthiazol-2-yl)–2,5-diphenyltetrazolium bromide (MTT) assay. α-Synuclein aggregates were prepared on a Corning 96-well Microplate. The fibrils were recovered, collected by centrifugation at 18,625 g for 20 min at room temperature and resuspended in fresh buffer (50 mM Tris-HCl, pH 7.4). Cells were seeded on 96-well plates (Greiner Bio-One) at 10000 cells/well in 100 μL medium. After 24 h at 37 °C, the medium was discarded and cells were treated in quintuplicates either with fresh medium (medium control), 20% (v/v) 50 mM Tris-HCl, pH 7.4, in fresh medium (buffer control), or α-synuclein aggregates at 20% (v/v) 50 mM Tris-HCl, pH 7.4, in fresh medium. After another 24 h at 37 °C, medium was discarded, and cells were incubated with MTT (Abcam) diluted 1:10 in RPMI medium (Thermo Fisher Scientific) for 4 h at 37 °C. The solution was discarded, and the formazan product was solubilised at 500 rpm and 37 °C for 15 min in 100 μL cell lysis buffer per well (Abcam) on a PHMP Grant-Bio Thermoshaker. Absorbance at 570 nm was measured on a CLARIOStar plate reader (BMG Labtech), and cell viability was calculated respective to medium control.

### *C. elegans* strains and maintenance

The *C. elegans* AM134 ((rmIs126[P(unc-54)Q0::YFP]), (YFP)) strain was the control strain used in this study, other strain used was, α-synuclein transgenic strain OW40 (zgIs15 [P(unc-54)αsyn::YFP]), in which α-synuclein is expressed in the body wall muscle cells and fused to YFP[8,41]. Synchronised axenic nematodes were acquired by bleaching gravid hermaphrodite adult nematodes with sodium hyochlorite. Axenised eggs were resuspended in minimal media buffer (M9) (0.3% KH2PO4, 0.6% Na2HPO4, 0.5% NaCl and 1 mM/L MgSO4), incubated overnight at 20 °C to hatch. Nematodes were maintained on OP50 *Escherichia coli* bacterial strain seeded on nematode growth medium (NGM) (0.3% NaCl, 0.75% casein, 1 mM MgSO4, 1 mM CaCl2, 250 mM KH2PO4 (pH 6) and 5 μg/mL mM cholesterol) agar (1.7%) plates. For experimental purposes, 5-fluoro-2'deoxyuridine (FUDR) (75 μM) were added to agar plates to inhibit generation of offspring. 1% of DMSO was seeded onto bacteria lawn of seeded plates for control and 5 μM of claramine was added to experimental plates. Experiments were conducted with L4 stage nematodes at 20 °C and plates were incubated at 20 °C.

### Confocal microscopy inclusions quantification in *C. elegans*

At indicated time points, nematodes were washed with M9, and palleted in 1 mL of M9 solution containing anaesthetic levamisole (10 mM) (Sigma) to induce paralysis. Nematodes were mounted onto a glass Petri plate (MatTek) with the aid of CyGEL™ (Biosatus, Ltd), a thermo-reversible hydrogel which is liquid when cold and a gel when warmed at room temperature. High-magnification images were acquired with a Leica TCS SP5 inverted confocal microscope scope with an 20x/1.3 HC PL Apo CS2 oil objective. YFP was detected using 512 nm as excitation and an emission range from 549–550 nm. At least 4 nematodes were imaged per condition. Representative confocal images of worms displaying the head (between the tip of nose and the pharyngeal bulb) were analysed using Fiji[47]. Inclusions were defined as having an area >10 square pixels (0.2 μm/pixel) and a circularity of 0.5-1.

### Automated *C. elegans* motility assay

L4 nematodes were cultured at 20 °C on either DMSO (1%) or claramine (5 μM), motility assay was carried out on NGM agar plates, at indicated time points, and the nematodes were washed off the plates with M9 buffer and spread over an unseeded NGM agar plate, after which their movements were recorded at 20 fps, using a lab-developed microscopic setup between 30 s to 1 min. At least 100 nematodes were counted in each experiment unless stated otherwise. Videos were analysed using a custom-made tracking code.

### *C. elegans* fluorescence recovery after photobleaching (FRAP)

FRAP experiments were performed on nematodes at indicated time-points using a Leica Stellaris Will inverted stage scanning confocal microscope. To conduct FRAP experiments, a 63x magnification oil objective (63x/1.4 HC PL Apo CS oil) was used. Nematodes were paralysed and immobilised using levamisole (10 mM) and CyGEL respectively on a glass bottom petri plate. Bleaching was done using the 488 nm laser at 20% intensity for 5 s following a 2 s pre-bleach sequence. Immediate post-bleach images were captured at ~1300 ms per frame rate, for ~25 s. Intensity traces of bleached area were background corrected and normalised to reference signal and FRAP time which is defined by the time to half maximal signal recovery. This was determined using the Leica TCS Stellaris FRAP wizard system on the confocal microscope.

### Statistics and reproducibility

Figure 1B displays representative images illustrating condensate formation under varying PEG and α-synuclein concentrations under both control and claramine conditions. This experiment was conducted at least three times to ensure reproducibility (the corresponding data for Fig. 1D, E, as well as Supplementary Fig. 2, can be found in the source file). This also applies to Supplementary Fig. 3B, where the replicates are detailed in the source file for Supplementary Fig. 3C. Additionally, the experiment presented in Fig. 2D was replicated four times, aligning with the experiments depicted in Fig. 2B, C. Samples were collected after each biological replicate, consistently yielding similar results. In summary, all experiments in this study was repeated at least three times.

### Reporting summary

Further information on research design is available in the Nature Portfolio Reporting Summary linked to this article.

## Data availability

Additional data generated in this study are provided in the Supplementary Information and source data files. Source data are provided with this paper.

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

## Acknowledgements

We would like to acknowledge Roman Misteli for assistance with ChemDraw chemical structures. S.T.D acknowledges funding from the Biotechnology and Biological Sciences Research Council (BBSRC; BB/M011194/1). Z.T. acknowledges funding from the Ron Thomson Research Fellowship in Alzheimer's Disease, Pembroke College, Cambridge. A.R. acknowledges funding from the European Union's Horizon 2020 research and innovation programme under the Marie Skłodowska-Curie grant agreement No 956977. M.V. acknowledges funding from UKRI (10059436 and 10061100).

## Author contributions

S.T.D, Z.T. and M.V. conceived the project. S.T.D, Z.T., M.P.C., A.R., M.C.H., O.M.M. and L.K.M. performed experiments. All authors analyzed data and wrote the article.

## Competing interests

The authors declare no competing interests.
