## [Peer Review File · Nature Communications]

Reviewers' Comments:

Reviewer #1:

Remarks to the Author:

Manuscript#: NCOMMS-23-38333-T

The manuscript entitled "Pharmacological inhibition of α -synuclein aggregation within liquid condensates" by Dada et al reported the effects of amino sterol claramine on α -synuclein aggregation. They showed that Claramine promotes condensation of α -synuclein by LLPS. Further they reported that Claramine stabilizes α -synuclein condensates, and inhibits its aggregation both in vitro and in a *C. elegans* model of Parkinson's disease (PD). Claramine also found to reduce α -synuclein aggregation, improve motility and fitness during aging in the worms. This work provides significant contribution to the field. The methods and rationale of the study seems sound and the work supports the conclusions. However, there are certain concerns which need to be addressed to consider this manuscript for publication in Nature Communications.

Major Comments

1. How the α -synuclein aggregation observed within condensates differ from primary-nucleation and growth-based strategies? Are these two processes mutually exclusive? From the manuscript it is evident that the aggregation within condensates also follow the nucleation and growth-based strategies. The distinction should be more clear or else it just does not make any difference in novelty of this study whether the Claramine is exclusively hindering the aggregation within condensates. It is important for the authors to check because a recent study (Srivastava et al, Journal of Neurochemistry 2023), showed the Shikonin based α -synuclein aggregation inhibition which functions approximately the same way like Claramine in reducing aggregate load, delay in lag phase, favouring helical structures without involving any statement on condensates. These observations also hold true for Claramine mediated aggregation reduction. Did the preformed fibrils also been generated through these condensation-based pathway?
2. Mass spectrometry data suggests Claramine does not covalently modifies α -synuclein but stabilizes it. What is the mechanism of the condensate stabilization? Please provide a clear picture of the mode of interaction between Claramine and α -synuclein.
3. How long did it exactly take for these fibrils to form (Figure-2D). Did these fibrils mature from the condensate? In both DMSO control and Claramine fibrils are present, are they toxic? How does the condensate stabilization by Claramine has anything to do with the endpoint of this aggregation reaction? Does the reduction in fibrils suggests some disaggregation within condensates?
4. How did the authors choose the dose for in-vivo experiment in *C. elegans*? Will there be a disparity in the in-vitro experiments if 5 μ M dose of Claramine would have been used? Would it be toxic for the animals at higher doses used in in-vitro experiments? Does the overall solubility of α -synuclein in-vivo changes in the presence of Claramine or induce more disaggregation of α -synuclein inclusions?
5. The experiments only have been performed using fluorescence microscopy. To further support, some biochemical experiments like western blotting for soluble and insoluble fractions might help.
6. Please supplement the manuscript with muscle architecture when muscle specific α -synuclein is expressed? Does musculature is affected in presence of α -synuclein and whether Claramine ameliorates the condition?
7. Although invertebrate model suffices the action of Claramine in ameliorating aggregation and improving motility, the effect of Claramine on mammalian cell lines would be a better way forward to show the efficacy and its suitability for the drug candidate.

Minor comments

Figures quality must be improved and typos should be corrected.

1. In Fig 1D,1E, and Supp Fig1, there are irregular boxes are present on y-axis. They probably used text boxes to hide the text numbers and the outline is still present.

2. On page 5 (7th line) YPF instead of YFP. On page 5 (Supple) Beach instead of Bleach.

3. The control strain used AM134 as reported in CGC "Strain was reported as incorrect; apparently carries 20Q transgene" Although the strain is still suitable but pure YFP constructs would have been a better control.

4. Please comment what would differ if WT α -synuclein would have been labelled with Alexa-fluor647. Does the addition of this cysteine variant affect condensate formation?

5. Discussion should be more elaborative.

Reviewer #2:

Remarks to the Author:

In this manuscript, the authors investigated the impact of claramine on the liquid-liquid phase separation and amyloid aggregation of α -synuclein in vitro and in vivo. They claimed that claramine is capable of reducing the threshold for liquid-liquid phase separation of α -synuclein and inhibiting the formation of fibrils within condensates. Their data also demonstrated that claramine suppresses the amyloid aggregation of α -synuclein within condensates by controlling the rate of primary nucleation while having a mild effect on secondary nucleation and elongation. Finally, the authors observed that claramine can maintain the liquid state of inclusions and restore the locomotor ability in a *C. elegans* model of PD. The work is potentially interesting but is a little premature, and should be improved by addressing the following.

Major Comments:

1.The buffer in the in vitro experiment was "physiological pH using polyethylene glycol (PEG)", how about the ionic strength? Previous reports have showed that the ionic strength is important to the formation of α -synuclein. The effect of claramine on α -synuclein LLPS in the physiological ionic conditions, that is around 100-150 mM NaCl/KCl, is necessary to be included.

2.It will be informational if claramine can be shown together with α -synuclein in the condensates.

3.In Figure 2B, the addition of 75 μ M claramine resulted in noticeable differences in the morphology of the α -synuclein condensates at 10-30 minutes compared to those without claramine. FRAP experiment is recommended to investigate the mobility properties of α -synuclein condensates after incubating for 30 minutes.

4.The ThT fluorescence intensity reduction in Figure 2 and 3 was used to demonstrate the inhibitory effect of claramine on the amyloid aggregation of α -synuclein. The control experiments are required to confirm whether claramine itself decreases the ThT fluorescence intensity and if the ThT fluorescence change is due to the probe displacement from α -synuclein.

5.The TEM of preformed fibrils used in Figure 3 need to be shown

6.The authors claimed that "claramine reduced the β -sheet formation of α -synuclein observed by the low ThT signal observed in comparison to the control". Here, a time-dependent CD measurement for α -synuclein condensates in the presence and absence of claramine is necessary to demonstrate if the transition of β -sheet secondary structure is reduced by claramine.

7.The authors claimed that "Claramine, therefore, stabilizes the condensate state of α -synuclein, while not modifying it covalently". The molecular mechanisms underlying the impact of claramine on α -synuclein liquid-liquid phase separation require further investigation. For instance, it is necessary to elucidate the specific interactions between claramine and α -synuclein.

8.The author investigated the mobility properties of α -synuclein inclusions in a *C. elegans* model of Parkinson's disease using FRAP experiments, demonstrating that claramine can maintain the mobility of inclusions. There seems to be a confusion here regarding whether α -synuclein inclusions and condensates are the same. Are inclusions formed through liquid-liquid phase separation?

9.Several recent studies on the regulation of α -synuclein LLPS and amyloid aggregation in the

condensates by small molecules were reported. There are also reports showing the small molecules regulated LLPS of other amyloid proteins, like tau. More related publications should be cited and discussed will improve the novelty of this manuscript.

10. Whether the Protein labeling with A90C mutation affects the LLPS of α -synuclein needs to be confirmed.

11. The authors discussed trodusquemine and squalamine in the manuscript. How about these two? Do they have a similar effect on α -synuclein LLPS?

Reviewer #1

The manuscript entitled “Pharmacological inhibition of α -synuclein aggregation within liquid condensates” by Dada et al reported the effects of amino sterol claramine on α -synuclein aggregation. They showed that Claramine promotes condensation of α -synuclein by LLPS. Further they reported that Claramine stabilizes α -synuclein condensates, and inhibits its aggregation both in vitro and in a *C. elegans* model of Parkinson's disease (PD). Claramine also found to reduce α -synuclein aggregation, improve motility and fitness during aging in the worms. This work provides significant contribution to the field. The methods and rationale of the study seems sound and the work supports the conclusions. However, there are certain concerns which need to be addressed to consider this manuscript for publication in Nature Communications.

We are grateful to the reviewer for acknowledging that “*this work provides significant contribution to the field*”, and for suggesting improvements of the presentation.

Major Comments

1. How the α -synuclein aggregation observed within condensates differ from primary-nucleation and growth-based strategies? Are these two processes mutually exclusive? From the manuscript it is evident that the aggregation within condensates also follow the nucleation and growth-based strategies. The distinction should be more clear or else it just does not make any difference in novelty of this study whether the Claramine is exclusively hindering the aggregation within condensates. It is important for the authors to check because a recent study (Srivastava et al, Journal of Neurochemistry 2023), showed the Shikonin based α -synuclein aggregation inhibition which functions approximately the same way like Claramine in reducing aggregate load, delay in lag phase, favouring helical structures without involving any statement on condensates. These observations also hold true for Claramine mediated aggregation reduction. Did the preformed fibrils also been generated through these condensation-based pathway?

In our study, we produced preformed fibrils from bulk solution, allowing for precise quantification of the concentration. To clarify this point, we incorporated TEM images of the pre-formed fibrils into Figure 2D.

The overall topology of the kinetic network describing the aggregation mechanism of α -synuclein is the same along the deposition and condensation pathways, as we originally described in Dada et al. PNAS 2023.

In the revised version of the manuscript (Introduction and Figure S1), we have now described how compounds that inhibit α -synuclein aggregation along the deposition pathway (such as Shikonin, now reference 30) may not necessarily inhibit α -synuclein aggregation along the condensation pathway, since the increase in thousand-fold concentration in the condensates requires very potent compounds to observe an inhibitory effect.

2. Mass spectrometry data suggests Claramine does not covalently modifies α -synuclein but stabilizes it. What is the mechanism of the condensate stabilization?

Please provide a clear picture of the mode of interaction between Claramine and α -synuclein.

To address this point, we have now carried out microscale thermophoresis (MST) experiments. The MST assay reveals that claramine does not covalently bind and exhibits weak binding to α -synuclein (K_D 20 μ M). MST data has been included in Supplementary Figure 4C-D.

3. How long did it exactly take for these fibrils to form (Figure-2D). Did these fibrils mature from the condensate? In both DMSO control and Claramine fibrils are present, are they toxic? How does the condensate stabilization by Claramine has anything to do with the endpoint of this aggregation reaction? Does the reduction in fibrils suggests some disaggregation within condensates?

Our experiments indicate that the fibrils are formed within the condensates (Dada et al. PNAS 2023).

We directly extracted fibrils from the aggregation assay within the condensates immediately after the 30-minute time point and washed with 50 mM Tris-HCL buffer. In Supplementary Figure 9B, we performed a cell toxicity assay using these fibrils, both in the presence and absence of claramine, on human neuroblastoma SH-SY5Y cells.

The lack of reduction in ThT signal over time, as well as the results from the new quenching assay data in Supplementary Figure 6B, indicate that there is no evident disaggregation of fibrils within condensates by claramine.

4. How did the authors choose the dose for in-vivo experiment in *C. elegans*? Will there be a disparity in the in-vitro experiments if 5 μ M dose of Claramine would have been used? Would it be toxic for the animals at higher doses used in in-vitro experiments? Does the overall solubility of α -synuclein in-vivo changes in the presence of Claramine or induce more disaggregation of α -synuclein inclusions?

The selected concentration was based on the observed toxicity of claramine at higher concentrations, as demonstrated in our experiments with human neuroblastoma SH-SY5Y cells (Supplementary Figure 9A).

5. The experiments only have been performed using fluorescence microscopy. To further support, some biochemical experiments like western blotting for soluble and insoluble fractions might help.

These experiments are detailed in Supplementary Figure 7.

6. Please supplement the manuscript with muscle architecture when muscle specific α -synuclein is expressed? Does musculature is affected in presence of α -synuclein and whether Claramine ameliorates the condition?

As we focus primarily on the investigation of the influence of claramine on α -synuclein aggregation, we would consider the physiological effects on the muscle architecture of *C. elegans* as outside the scope of our work.

7. Although invertebrate model suffices the action of Claramine in ameliorating aggregation and improving motility, the effect of Claramine on mammalian cell lines would be a better way forward to show the efficacy and its suitability for the drug candidate.

We conducted these experiments on human neuroblastoma SH-SY5Y cells, as illustrated in Supplementary Figure 9.

Minor comments

Figures quality must be improved and typos should be corrected.

1. In Fig 1D,1E, and Supp Fig1, there are irregular boxes are present on y-axis. They probably used text boxes to hide the text numbers and the outline is still present.

We have now addressed this point.

2. On page 5 (7th line) YPF instead of YFP. On page 5 (Supple) Beach instead of Bleach.

We have now addressed this point.

3. The control strain used AM134 as reported in CGC "Strain was reported as incorrect; apparently carries 20Q transgene" Although the strain is still suitable but pure YFP constructs would have been a better control.

We are grateful to the referee for pointing this out, which we will consider in future studies.

4. Please comment what would differ if WT α -synuclein would have been labelled with Alexa-fluor647. Does the addition of this cysteine variant affect condensate formation?

It was demonstrated that the A90C variant is virtually identical to the unlabeled wild-type (WT) protein (Cremades et al. Cell 149, 1048-1059, 2012). Additionally, only 1% of the labeled protein was introduced into the assay.

5. Discussion should be more elaborative.

We hope that the additional discussions that we included in the revised version of the manuscript is helpful.

Reviewer #2

In this manuscript, the authors investigated the impact of claramine on the liquid-liquid phase separation and amyloid aggregation of α -synuclein in vitro and in vivo. They claimed that claramine is capable of reducing the threshold for liquid-liquid phase separation of α -synuclein and inhibiting the formation of fibrils within condensates. Their data also demonstrated that claramine suppresses the amyloid aggregation of α -synuclein within condensates by controlling the rate of primary nucleation while having a mild effect on secondary nucleation and elongation. Finally, the authors observed that claramine can maintain the liquid state of inclusions and restore the locomotor ability in a *C. elegans* model of PD. The work is potentially interesting but is a little premature, and should be improved by addressing the following.

We would like to thank the reviewer for the overall positive assessment and for providing a series of helpful suggestions on how to strengthen our conclusions.

Major Comments:

1. The buffer in the in vitro experiment was “physiological pH using polyethylene glycol (PEG)”, how about the ionic strength? Previous reports have showed that the ionic strength is important to the formation of α -synuclein. The effect of claramine on α -synuclein LLPS in the physiological ionic conditions, that is around 100-150 mM NaCl/KCl, is necessary to be included.

We have now carried out additional experiments to investigate the effects of ionic strength, and reported them in Supplementary Figure 5. In the revised version of the manuscript, we have also noted that the phase separation propensity of α -synuclein is adversely affected in a high ionic environment, as demonstrated in previous studies (e.g. reference 26).

2. It will be informational if claramine can be shown together with α -synuclein in the condensates.

We have been thinking about how to address this point. However, claramine lacks an intrinsic fluorophore within the visible wavelength, as shown in Supplementary Figure 5A. Alternatively, conjugating a tag to claramine would alter significantly its structure and effects. Since it has been challenging to design an experiment to respond to this question, we would welcome suggestions to this effect.

3. In Figure 2B, the addition of 75 μ M claramine resulted in noticeable differences in the morphology of the α -synuclein condensates at 10-30 minutes compared to those without claramine. FRAP experiment is recommended to investigate the mobility properties of α -synuclein condensates after incubating for 30 minutes.

At 30 minutes the aggregation of α -synuclein has already taken place (Figure 2C), so FRAP experiments at this time point may not provide significant additional information.

4. The ThT fluorescence intensity reduction in Figure 2 and 3 was used to demonstrate the inhibitory effect of claramine on the amyloid aggregation of α -synuclein. The control experiments are required to confirm whether claramine itself decreases the ThT fluorescence intensity and if the ThT fluorescence change is due to the probe displacement from α -synuclein.

A fluorescence quenching assay has now been carried out, as shown in Supplementary Figure 6B.

5. The TEM of preformed fibrils used in Figure 3 need to be shown.

These images have now been added to Figure 2D.

6. The authors claimed that “claramine reduced the β -sheet formation of α -synuclein observed by the low ThT signal observed in comparison to the control”. Here, a time-dependent CD measurement for α -synuclein condensates in the presence and absence of claramine is necessary to demonstrate if the transition of β -sheet secondary structure is reduced by claramine.

FTIR analysis was performed as illustrated in Figure 2E-F. Since CD analysis demands a substantial amount of sample material, which is impractical for our assay, we conducted biochemical characterisation on the soluble and insoluble fractions to investigate structural variations in α -synuclein in the presence and absence of claramine shown in Supplementary Figure 7.

7. The authors claimed that “Claramine, therefore, stabilizes the condensate state of α -synuclein, while not modifying it covalently”. The molecular mechanisms underlying the impact of claramine on α -synuclein liquid-liquid phase separation require further investigation. For instance, it is necessary to elucidate the specific interactions between claramine and α -synuclein.

The MST binding assay was conducted, and the results are presented in Supplementary Figure 4C-D.

8. The author investigated the mobility properties of α -synuclein inclusions in a *C. elegans* model of Parkinson's disease using FRAP experiments, demonstrating that claramine can maintain the mobility of inclusions. There seems to be a confusion here regarding whether α -synuclein inclusions and condensates are the same. Are inclusions formed through liquid-liquid phase separation?

In a previous study (reference 26), we showed that the inclusions are indeed condensates. In that study, the material properties were modulated by hexanediol. We have employed the same model in this study.

9. Several recent studies on the regulation of α -synuclein LLPS and amyloid

aggregation in the condensates by small molecules were reported. There are also reports showing the small molecules regulated LLPS of other amyloid proteins, like tau. More related publications should be cited and discussed will improve the novelty of this manuscript.

We are grateful to the referee for making this point. We have now added reference to a recent study on tau (references 29-31).

10. Whether the Protein labeling with A90C mutation affects the LLPS of α -synuclein needs to be confirmed.

It has already been reported (Cremades et al. Cell 149, 1048-1059, 2012) that the A90C variant is virtually identical to the unlabeled wild-type (WT) protein. Additionally, only 1% of the labeled protein was introduced into the aggregation assay.

11. The authors discussed trodusquemine and squalamine in the manuscript. How about these two? Do they have a similar effect on α -synuclein LLPS?

As we now report in Supplementary Figure 3, spermine, the principal functional group in all these aminosterols, does not exert the same influence on α -synuclein phase separation as claramine. This finding underscores the importance of carrying out detailed kinetic studies for each individual compound, which we plan to do in future studies for squalamine and trodusquemine.

Reviewers' Comments:

Reviewer #1:

Remarks to the Author:

The manuscript entitled "Pharmacological inhibition of α -synuclein aggregation within liquid condensates" by Dada et al reported the effects of amino sterol claramine on α -synuclein aggregation. They showed that Claramine promotes condensation of α -synuclein by LLPS. Further they reported that Claramine stabilizes α -synuclein condensates, and inhibits its aggregation both in vitro and in in a *C. elegans* model of Parkinsons disease (PD). Claramine also found to reduce α -synuclein aggregation, improve motility and fitness during aging in the worms. The methods and rationale of the study seems sound and the work supports the conclusions.

The authors have addressed the previous comments. They have added more references on claramine-like compounds like squalamine and trodusquemine. They have incorporated how the compounds that inhibit α -synuclein aggregation along the deposition pathway work. They have included the MST data, the quenching assay data, the insoluble fraction western blot data. The authors have now described the whole story in detail. They have communicated about why they shortlisted claramine, how claramine stabilizes condensates, how it inhibits aggregation, and how it contributes to the fitness of PD models. However, it remains elusive how the α -synuclein aggregation chooses between the deposition and condensation pathway and the effect of the aggregates in vivo from these two pathways. Although these are beyond the scope of this work, this establishes a crucial small molecule as a potent inhibitor of α -synuclein and thus a possible drug option for Parkinson's disease.

Over all, this study significantly contributes to the field, and is recommended for publication.

Reviewer #2:

Remarks to the Author:

The authors have addressed most of my concerns. However, I think the FRAP experiments are necessary to verify the effect of claramine on the fluidity of α -synuclein condensates in Figure 2B (see last round my comment 3). At ten minutes, the condensates of α -synuclein have undergone irregular deformation in the presence of 75 μ M claramine, but the ThT data indicated that amyloid aggregation had not occurred yet. The authors need to provide FRAP or other experiments to explain why claramine deforms the condensates of α -synuclein.

Reviewer #2

The authors have addressed most of my concerns. However, I think the FRAP experiments are necessary to verify the effect of claramine on the fluidity of α -synuclein condensates in Figure 2B (see last round my comment 3). At ten minutes, the condensates of α -synuclein have undergone irregular deformation in the presence of 75 μ M claramine, but the ThT data indicated that amyloid aggregation had not occurred yet. The authors need to provide FRAP or other experiments to explain why claramine deforms the condensates of α -synuclein.

We have now carried out the FRAP experiments requested. The results indicate that claramine helps maintain the alpha-synuclein condensate in a liquid-like state (Supplementary Figure 6). These data were acquired at 15 minutes, since the difference between the presence and absence of claramine is more evident at this time than at 10 minutes (see figure below), when both samples are still liquid-like.

Thus, given the absence of at ThT fluorescence signal and the recovery of Alexa-647 fluorescence after photobleaching, the irregular shape of the condensates in the presence of claramine at 10 minutes can be attributed to the wetting of a surface with small irregularities.